# Yeasts as a Potential Biological Agent in Plant Disease Protection and Yield Improvement—A Short Review

Jolanta Kowalska [1],* , Joanna Krzymińska [1] and Józef Tyburski [2]

1   Department of Organic Agriculture and Environmental Protection,
    Institute of Plant Protection—National Research Institute, Władysława Węgorka 20, 60-318 Poznan, Poland
2   Department of Agroecosystems and Horticulture, University of Warmia and Mazury,
    Michala Oczapowskiego 2, 10-719 Olsztyn, Poland
*   Correspondence: j.kowalska@iorpib.poznan.pl

**Abstract:** The role of biocontrol products is expected to increase worldwide consumer demand and facilitate the implementation of sustainable agricultural policies. New biocontrol agents must allow for an effective crop-protection strategy in sustainable agriculture. Yeasts are microorganisms living in various niches of the environment that can be antagonists of many plant pathogens. Yeasts rapidly colonize plant surfaces, use nutrients from many sources, survive in a relatively wide temperature range, produce no harmful metabolites and have no deleterious effects on the final food products. Hence, they can be a good biocontrol agent. In this paper, the biological characteristics and potential of yeast are summarized. Additionally, the mechanisms of yeasts as plant-protection agents are presented. This includes the production of volatile organic compounds, production of killer toxins, competition for space and nutrient compounds, production of lytic enzymes, induction of plant immunity and mycoparasitism. The mechanisms of yeast interaction with plant hosts are also described, and examples of yeasts used for pre- and postharvest biocontrol are provided. Commercially available yeast-based products are listed and challenges for yeast-based products are described.

**Keywords:** biological agent of protection; microbial antagonism; enzyme secretion; organic agriculture

## 1. Introductions

The role of bioproducts in agricultural practices world-wide—biopesticides and biofertilizers, i.e., "microbial-based" pesticides or fertilizers—is expected to increase in an effort to implement sustainable agriculture policies. Many microbial strains have the capacity to augment plant productivity by enhancing crop nutrition and functioning as biopesticides [1]. This includes yeasts, which affect harmful microorganisms both directly and indirectly.

Crops, both during their growth and after harvest, are exposed to the negative impact of pathogenic microorganisms. Plant pathogens significantly impair the quantity and quality of crops and their potential suitability for the consumer. Infected products can also pose a health risk. In intensive agriculture, chemical plant-protection products are widely used. However, the use of chemical plant-protection products in organic farming is prohibited. Additionally, their use carries risks such as the increasing resistance of pathogens to the active substances, which reduces pesticide efficacy. Their use also causes the deposition and accumulation of pesticide active substances in soil and water, which in general alters the functioning of ecosystems, and in particular disrupts the interactions between organisms. Therefore, there is a need to look for alternative methods of plant protection in relation to organic farming systems and sustainability systems.

Nowadays, consumers are increasingly considering the impact of how food is produced based on ethical and pro-environmental motives, and are opting for organically grown products. Biopesticides based on microorganisms, including yeasts, can help to provide such products. Despite the fact that biocontrol agents have been researched for

more than fifty years, which has led to the commercial registration of biological agents, new biocontrol agents still need to be introduced into the biopesticide market; this will allow an effective crop-protection strategy in sustainable agriculture. In particular, the use of antagonistic yeasts as potential biocontrol agents still needs to be explored and brought into wider and more common use. Among the microorganisms that are potential antagonists to plant pathogens, yeast meets all the conditions for an effective antagonist against plant disease agents: they must rapidly colonize the plant surface, have the ability to utilize components from a variety of sources, survive in a relatively wide temperature range, produce no harmful metabolites and have no deleterious effects on the final food product. Their metabolic activity is evident. They thrive in many ecosystems, where they interact with other microorganisms, including reducing the abundance of phytopathogens in the environment. They can be cultured relatively easily and their metabolites can be collected, which makes it possible to select those yeast species and strains which can potentially be used as biofungicides.

Yeasts are microscopic eukaryotic (i.e., living organisms having a cell nucleus surrounded by a nuclear membrane) organisms in the kingdom of Fungi (mainly Ascomycota and Basidiomycota phyla). As heterotrophic organisms, they require carbon and nitrogen as nutrients. Under aerobic conditions, yeasts assimilate carbohydrates to produce $CO_2$ and $H_2O$, and under anaerobic conditions, they convert carbohydrates to alcohol through the process of fermentation. Their cells (round, ellipsoidal, oval, or cylindrical) form colonies. Their shape and size depend on the species, the condition of the culture and the age of the colony. The cells typically range in size from 3 to 10 μm in length and 2 to 7 μm in width. They can be haploid, diploid, or polyploid. They are either homothallic (the exchange of genetic material occurs in one individual) or heterothallic (another individual is required to exchange genetic material). Their morphology as single-cell organisms allows for adhesion and biofilm formation. This impacts their ability to survive in the environment and enhances their competitiveness.

They reproduce vegetatively (asexually) and generatively (sexually). The first kind of asexual reproduction in yeast—budding—requires the right environmental conditions, which include temperature and availability of nutrients. This type of reproduction is characteristic of *Candida, Saccharomyces*, *Pichia* and *Rhodotorula*. The buds (daughter cells) are identical to the parent cells but smaller. The cells separate from the parent cell and form a separate organism, or—as in the case of the genus Candida—fuse with it and form a pseudomycelium. The second type of asexual reproduction is fission. In this process, the cell grows by elongating in one direction and daughter cells are identical in size to the parent cell. This type of reproduction is characteristic of *Schizosaccharomyces*. Under stress conditions such as lack of nutrients, yeasts undergo sporulation. The shape of the spores is characteristic of each yeast species. During sexual reproduction, haploid spores conjugate, forming diploids.

Their complex genome organization reduces the frequency of horizontal gene transfer, compared with other fungi. Additionally, most yeast species (excluding many *S. cerevisiae* strains) lack plasmids, which excludes the risk of plasmid-based pathogenicity and toxin biosynthesis genes.

## 2. Bioactivity Mechanisms of Yeasts

The effects of yeasts on plants and their pathogens are not as comprehensively described as other microorganisms, such as bacteria or filamentous fungi. They have positive effects on the growth and protection of crop plants, both directly and indirectly. They have positive effects on plants both as biostimulants and as biopesticides, reducing the development and impact of pathogens. To successfully apply yeasts as plant-protection agents, it is crucial to understand the mechanisms underlying their interaction with plants and plant pathogens.

For the biocontrol of plant pathogens using yeasts, multiple mechanisms such as the production of volatile organic compounds, production of toxins, competition for space

and nutrient compounds, production of lytic enzymes, induction of plant immunity and mycoparasitism are involved (Table 1). In most yeast species, a few mechanisms occur simultaneously, which enhance the antagonistic effect against phytopathogens.

Volatile organic compounds (VOCs) are produced by microorganisms such as fungi, bacteria and yeast during their primary and secondary metabolism [2]. They are species-specific and are involved in intercellular communication, as well as in supporting or limiting the growth of other microorganisms [3]. Their chemical composition (a blend of volatiles called volatilome) strongly depends on the environment and the pathogen being antagonized [4]; the chemical composition includes alcohols, aldehydes, cyclohexanes, benzene derivatives, heterocyclic compounds, hydrocarbons, ketones, phenols, thioalcohols and thioesters. These are small molecules (generally under 300 Da) poorly soluble in water and with a high vapor pressure at room temperature. Physical contact between the biocontrol agent and the pathogen is not needed. The role of volatilome has been described in recent studies. VOCs are produced by such species as *Sporidiobolus pararoseus* Fell & Tallman [5], *Candida sake* [6], Hanseniaspora [7], *Wickerhamomyces anomalus* (E.C. Hansen) Kurtzman, *M. pulcherrima*, *Aureobasidium pullulans* and *S. cerevisiae* [8,9]. They have been proven to successfully reduce the growth of such pathogens as *B. cinerea*, *Colletotrichum acutatum* J.H. Simmonds, *Penicillium expansum* Link, *Penicillium digitatum* [Pers.] Sacc. and *Penicillium italicum* Wehmer [5,9,10].

Yeasts typically do not produce any toxic metabolites, which makes them biologically safe. In 2016, the *Saccharomyces cerevisiae* strain LAS02 was approved as a low-risk active substance under Regulation (EC) No 1107/2009 of the European Parliament and of the Council concerning the placing of plant-protection products on the market, and amending the Annex to Commission Implementing Regulation (EU) No 540/2011. Some strains of yeast can produce toxic proteins that cause degradation of the cell membrane of selected organisms [11]. These killer yeasts are resistant to their own toxins. This mechanism was initially discovered in *S. cerevisiae* [12] and is toxic to other yeast species; however, the toxins have shown activity against other species and types of microorganisms, including bacteria and filamentous fungi [11–14]. This mechanism was then reported for other yeast species: *Pseudozyma flocculosa* (Traquair, L.A. Shaw & Jarvis) Boekhout & Traquair [15], *Pichia anomala* E.C. Hansen [16], *Wickerhamomyces* sp. [17], *Pichia membranifaciens* E.C. Hansen [18], *Debaryomyces hansenii* Zopf [13], *Kluyveromyces lactis* (Stell.-Dekk.) Van der Wal [19] and *A. pullulans* [20]. In biological plant protection, toxins produced by yeast can be used in order to obtain products safe for human consumption and environment. Modes of yeast action are varied, even with respect to the same species; these various modes are presented in Table 2.

Yeasts, like all microorganisms, compete with other organisms, including plant pathogens, for nutrients and space [21,22]. This mechanism, which is considered their primary mode of action, is important when protecting plant products in storage, including fruit storage [21], as well as in the natural environment, where resources are limited. Yeasts grow rapidly and intensively, forming a biofilm on the surface of plants—a membrane of interconnected microorganisms, which can be treated as a single organism or a consortium, and which cause inhibition of the pathogen's mycelial growth and spore production [22]. Additionally, yeasts colonize plant surfaces, especially in damaged areas, where the plant is most prone to infection by pathogens that access released nutrient substrates [23]. Yeasts deplete the available nutrients to build their biomass and thus limit nutrient availability to pathogens. During biofilm formation, individual yeast cells attach themselves to the plant surface and form an intercellular network, as well as hyphae and pseudohyphae [24,25]. Biofilms are formed by species such as *Pichia fermentans* Lodder, *A. pullulans*, *Kloeckera apiculata* (Niehaus) Shehata, Mrak & Phaff ex M.T. Sm, *S. cerevisiae*, *Pichia kudriavzevii* Boidin, Pignal & Besson, *W. anomalus* and *M. pulcherrima* [26–30]. Competition for nutrients can result in complete inhibition of pathogenic fungal spore germination. For example, studies have shown inhibition of growth of *P. expansum* [31] and *Monilinia laxa* (Aderh & Ruhland)

Honey [32] by *A. pullulans*. This interaction is attributed, among other things, to intense competition for iron, one of the key elements for microorganisms [22].

The production of lytic enzymes upon direct contact between the yeast and the pathogen is another mechanism that has been well-studied. This mechanism is particularly effective against necrotrophs [33]. Yeasts can secrete such enzymes as chitinases, glucanases, lipases, or proteases. Chitinases allow efficient degradation of the cell wall of plant pathogens, and their secretion is considered useful for biocontrol agents. Among yeasts, this activity has been described for such genera as *Candida*, *Metschnikowia*, *Meyerozyma*, *Pichia* and *Saccharomyces* [34–37]. Chitinases, moreover, can stimulate natural plant immune processes by degrading chitin and producing chitooligosaccharides [38]. Lipases are enzymes characterized by their interaction with non-water-soluble substrates. Their presence has been confirmed for such yeasts as genera *Candida* and *Cryptococcus* [39,40]. Beta-glucans are essential components of the fungal cell wall, responsible for cell adhesion and resistance to toxins. β-1,3-glucanase, which is produced by such yeasts as *Candida famata* (Zopf) Lodder & Kreger, *Rhodotorula mucilaginosa* (A. Jorg.) F.C. Harrison and *W. anomalus* [40–44], is effective in reducing pathogen growth.

Although proteases are recognized as an important factor in antagonistic processes, their production by yeast has not been thoroughly studied. Bar-Shimon et al. [34] described the secretion of proteases by *Candida oleophila* Montrocher, and Pretscher et al. [45] described the secretion of proteases by the genera *Metschnikowia*, *Pichia* and *Wickerhamomyces*.

Yeasts can also stimulate natural plant defence processes [46]. The plant's own immune system can recognize and respond to the presence of microorganisms, including pathogens. The resistance is induced systemically. Yeasts can induce the systemic defence of plants against many different pathogens by stimulating the production and activity of substances such as phytoalexins [47], chitinase and β-1,3-glucanase [48] and peroxidase [49].

Even though mycoparasitism (fungivory, where the fungus is consumed by another organism) is an important mechanism for plant protection, it is rather sporadically described. This mechanism is particularly effective, since yeast can adhere to the fungus cell wall, perforate it, and as a consequence stop the cell cycle, disrupting its morphology and lowering its turgor. This mechanism is linked to enzyme secretion such as glucanases (described above). *Pichia guilliermondii* (Wick.) Kurtzman & M. Suzuki [50]; *Pseudozyma aphidis* (Henninger & Windisch) Q.M. Wang, Begerow, F.Y. Bai & Boekhout [51]; *Saccharomycopsis schoenii* (Nadson & Krassiln.) Kurtzman & Robnett [52]; and *Vishniacozyma tephrensis* Vishniac ex Xin Zhan Liu, F.Y. Bai, M. Groenew. & Boekhout [53] have been shown to have mycoparasitic properties against plant pathogens.

**Table 1.** Multiple mechanisms of yeast as biocontrol agents—summary.

| Mechanism | Yeast Species |
|---|---|
| Volatile organic compound secretion | *Sporidiobolus pararoseus* [5], *Candida sake* [6], *Hanseniaspora* sp. [7], *Wickerhamomyces anomalus*, *Metschnikowia pulcherrima*, *Aureobasidium pullulans*, *S. cerevisiae* [8,9] |
| Toxic protein secretion | *Pseudozyma flocculosa* [14], *Pichia anomala* [15], *Wickerhamomyces* sp. [16], *Pichia membranifaciens* [17], *Debaryomyces hansenii* [18], *Kluyveromyces lactis* [19], *Aureobasidium pullulans* [19] |
| Competition for nutrients and space | *Pichia fermentans*, *Aureobasidium pullulans*, *Kloeckera apiculata*, *Pichia kudriavzevii*, *Wickerhamomyces anomalus*, *Metschnikowia pulcherrima* [25–30] |
| Lytic enzyme production | *Candida* sp., *Metschnikowia* sp., *Meyerozyma* sp., *Pichia* sp., *Saccharomyce* sp., *Rhodotorula mucilaginosa*, *Wickerhamomyces anomalus* [33–36,40–44] |
| Plant defence stimulation | *Pichia guilliermondii* [47], *Pseudozyma aphidis* [48] |
| Mycoparasitism | *Pichia guilliermondii* [49], *Pseudozyma aphidis* [50] *Saccharomycopsis schoenii* [51], *Vishniacozyma tephrensis* [52] |

**Table 2.** Killer toxins and their mode of action of *Pichia* spp. [18].

| *Pichia* Species | Toxin | Mode of Action |
|---|---|---|
| *P. acaciae* | NRRLY-18665 (PaT) | Cell cycle is stopped at G1, chitinase activity |
| *P. anomala* | NCYC434 (Panomycocin), ATCC 96603/K36/UP25F (PaKT), DBVPG 3003 (Pikt), YF07b (-),VKM-Y (WAKTa/b) | (1-3)-β-D-glucan hydrolysis |
| *P. farinosa* | KK1 (SMKT) | Permeabilization of membrane |
| *P. inositovora* | NRRL Y-18709 (-) | rRNA fragmentation |
| *P. kluyveri* | 1002 (-) | Permeabilization of membrane |
| *P. membranifaciens* | CYC 1106 (PMKT) <br> CYC 1086 (PMKT2) | Permeabilization of membrane, apoptosis <br> Cell cycle is stopped, apoptosis |
| *P. ohmeri* | 158 (-) | Loss of cellular integrity |

## 3. Interaction of Yeasts with Plant Hosts

Soil yeasts are mostly found in the rhizosphere [54–56] and have a positive effect on plant root growth [57–59]. Yeasts colonizing plants also improve their growth [60,61], including yeasts colonizing the leaf surface [62–64]. The mechanism for stimulating plant growth may involve making nutrients (such as nitrogen, phosphorus and potassium) available for plants [65]. They also produce plant hormones such as auxins or cytokinins that have a significant impact on the regulation of plant physiological processes and growth [66]. In addition, yeasts promote plant resistance to physiological stress [67].

Yeasts can increase nutrient availability to plants. Low nitrogen (N) availability is one of the most significant causes of reduced plant yield [68]. Among the microorganisms most commonly used for nitrogen acquisition by plants are bacteria [67]. However, selected yeast species also have this ability. These include species such as *Candida tropicalis* (Castell.) Berkhout isolated from soil [69], *Pseudozyma rugulosa* (Traquair, L.A. Shaw & Jarvis) Q.M. Wang, Begerow, F.Y. Bai & Boekhout, *Cryptococcus flavus* Saito and *Pseudozyma antarctica* (Goto, Sugiyama & Iizuka) Q.M. Wang, Begerow, F.Y. Bai & Boekhout [61,70]. The yeast-produced enzyme 1-aminocyclopropane-1-carboxylase (ACC) plays an important role in this process, causing the release of large amounts of ammonia, which triggers a microbe-mediated nitrogen-acquisition mechanism in plants [70]. Yeasts such as *Candida tropicalis* and various species of the genus *Cryptococcus* produce deaminases [71]. Other yeasts are involved in denitrification, which is the reduction of nitrate to nitrite or nitrogen during anaerobic respiration and converting it to biologically useful forms for plants [72–74]. After nitrogen, the second most important nutrient for plants is phosphorus (P) [75]. Its deficiencies can significantly affect crop yields. This element can be made available to plants by microorganisms from both organic and inorganic sources [76]. This includes *C. tropicalis* and *Lachancea thermotolerans* (Filippov) Kurtzman, which dissolve $Ca_3(PO_4)$, as well as the *Rhodotorula* genus, which provides dissolved phosphorus by lowering the pH [77–79]. Other soil-derived species, such as *Yarrowia lipolytica* (Wick., Kurtzman & Herman) Van der Walt & Arx and *S. cerevisiae*, dissolve inorganic phosphorus compounds via citric acid production [69]. The third of the most important macronutrients for plants is potassium (K), which plays an important role in many processes, including plant growth [70]. Since much of this element in the soil exists in the form of insoluble mineral compounds, the role of the microorganism in making it available is extremely important [80]. This includes species such as *Torulaspora globosa* Klöcker [56] and *Rhodotorula glutinis* (Fresen.) F.C. Harrison and *P. anomala* [81,82], which can significantly increase the availability of this element by lowering soil pH.

Other nutrient availability (calcium, iron, magnesium, sulfur and zinc) for plants can be increased by the presence of microorganisms. This process is usually caused by increasing the acidity of the rhizosphere through the production of organic acids. In the

literature, the yeasts *Williopsis californica* (Lodder) Kurtzman, Robnett & Basehoar-Power and *S. cerevisiae* are reported as oxidizing sulfur, as well as other nutrients [83–85].

Yeast can secrete phytohormones that have beneficial effects on plant growth. These phytohormones include such regulators as auxins containing indole, a heterocyclic chemical compound made of conjugated benzene and pyrrole rings [86], which regulates many important processes in plants [87]. The literature on auxin-producing yeasts mentions *R. paludigenum*, *S. cerevisiae*, *A. pullulans*, *Candida* sp., *Dothideomycetes* sp., *Hanseniaspora uvarum* (Niehaus) Shehata, Mrak & Phaff ex M.T. Sm., *Meyerozyma caribbica* (Vaughan-Mart. et al.) Kurtzman & Suzuki, *Meyerozyma guilliermondii* (Wickerham) Kurtzman & M. Suzuki, *Torulaspora* sp., *Barnettozyma californica* (Lodder) Kurtzman, Robnett & Basehoar-Power, *Cryptococcus laurentii* (Kuff.) X.Z. Liu, F.Y. Bai, M. Groenew. & Boekhout, *Rhodosporidiobolus fluvialis* (Fell, Kurtzman, Tallman & J.D. Buck) Q.M. Wang, F.Y. Bai, M. Groenew. & Boekhout *Candida maltosa* and *P. kudriavzevii* Komag., Nakase & Katsuya [58,66,70,88].

Another group of phytohormones synthesized by yeast is cytokinins. They have an important effect on the process of cell division in plants. Species producing them include *Sporobolomyces roseus* Kluyver & C. B. Niel, *M. pulcherrima* and *A. pullulans* [89]. Yeasts that produce gibberellic acid, a plant growth promoter that accelerates germination, can also be found [90].

Abiotic stress caused by adverse environmental conditions can significantly reduce crop yields, even by more than 50% [91]. However, this stress can be neutralized by the beneficial effects of microorganisms, including yeasts, by eliminating the effects of unfavorable temperatures [92,93], drought [94,95], salinity [96,97] or the presence of heavy metals [84]. One of the most commonly produced substances by plants in response to abiotic stress is the hormone ethylene [98–100]. This is a very effective plant regulator, which is active even at a low concentration. Each stage of plant development depends on its production [101]. However, excessive amounts of ethylene can be detrimental to the plant [102,103]. A deaminase enzyme located in the cytoplasm of microorganisms reduces the amount of ethylene and stimulates plant growth [104]. Yeasts such as *C. tropicalis*, *P. rugulosa*, *P. antarctica*, *A. pullulans*, *Dothideomycetes* sp., *Cryptococcus* sp., *R. paludigenum* and *T. globosa* have been reported to reduce ethylene and promote plant growth [58,67,105,106].

## 4. Use of Yeast for Preharvest Protection

Although yeasts are mainly used as antagonists against pathogens during the postharvest stage, examples of their successful preharvest use on wheat, vegetable crops and fruit have been described. It has been shown that a double treatment with *Rhodosporidium kratochvilovae* (Hamam., Sugiy. & Komag.] Q.M. Wang, F.Y. Bai, M. Groenew. & Boekhout), *C. laurentii* and *A. pullulans* combined with low doses of fungicides effectively reduced the infestation caused by powdery mildew of cereals and grasses by almost 90% and increased the yield and maturity of rice and/or wheat grain [73,107,108]. Wachowska and Glowacka [109] in a greenhouse experiment showed that a four-fold treatment with *A. pullulans* effectively stopped the development of *F. culmorum* pods on winter wheat and also caused fungi of the *Acremonium* and *Penicillium* genera to develop at a slower rate. Ponsone et al. [110] described the successful application of two strains of *Lanchancea thermotolerans* as biological antagonists against *Aspergillus niger* on grapes at harvest. The yeasts *Cryptococcus magnus* (Lodder & Kreger) Liu, Bai, Groenewald, & Boekhout, *Cryptococcus* sp., *S. pararoseus*, *A. pullulans* and *Rhodotorula* sp. effectively reduced symptoms of brown rot of stone fruit caused by *Monilinia fructicola* (G. Winter) Honey on nectarines [111].

Al-Ani et al. [112] observed a reduction in symptoms caused by potato virus Y after the application of yeast of the genus *Rhodotorula*, which also had a positive effect on germination, plant growth and dry weight.

## 5. Use of Yeast for Postharvest Protection

Biological control of postharvest diseases can be accomplished with commercially available products. Unfortunately, the availability of such products, especially those based on beneficial organisms whose activity greatly depends on environmental conditions, is still limited. Between 30 and 50 percent of fruit is lost during storage, never reaching the consumer [113]. A significant percentage of these losses is due to the effects of fungal pathogens such as *Alternaria*, *Botrytis*, *Colletotrichum*, *Fusarium*, *Monilia*, *Penicillium* and *Rhizopus*. It is important to minimize these losses, especially in the current era, with a growing population and dwindling natural resources [114]. Agents based on microorganisms, including yeast, are potentially useful in preventing such losses; however, their availability on the market is relatively limited. At the same time, consumer awareness and demand for high-quality products protected using natural methods are steadily increasing, so it seems important to research such biological agents.

Among the products protected by yeast-based agents, strawberries, grapes, tomatoes, apples, pears, mangoes and kiwis have been described. In the study by Kowalska et al. [115] the yeast species *Cryptococcus albidus* was evaluated for postharvest control of *Botrytis cinerea* in strawberries in two experiments. The percentage of decayed fruit increased after 10 days of storage, but was about 20% lower in the *C. albidus*-treated than in the untreated fruit.

One of the important pathogens in fruit storage is *B. cinerea*. Species such as *R. glutinis* [116], *Hanseniaspora opuntiae* Čadež, Poot, Raspor & M.T. Sm [117], *A. pullulans* [118,119] and *L. thermotolerans* and *M. pulcherrima* [120] were proven effective against grey mold. In other studies on protecting apples in storage, chitin isolated from the cell walls of *S. cerevisiae* was effective [121]. Studies on the effects of yeasts isolated from marine sediments show the effectiveness of *Scheffersomyces spartinae* (Ahearn, Yarrow & Meyers) Kurtzman & M. Suzuki and *Candida pseudolambica* M.T. Sm. & Poot in apple protection [122].

The genus *Aspergillus* is also a cause of losses in fruit storage. Yeasts from the genera *Rhodotorula*, *Metschnikowia*, *Saccharomyces* and *Pichia* were researched by Tryfinopoulou et al. [123] and were proven to be effective against *Aspergillus*. Li et al. [124] described the antagonistic effect of *S. pararoseus* against *Aspergillus niger* Tiegh. Jaibangyang et al. [125] described *Candida nivariensis* Alcoba-Flórez, Méndez-Álv., Cano, Guarro, Pérez-Roth & Arévalo as being effective against *Aspergillus flavus* Link.

Penicillium is also another storage-relevant pathogen. Assaf et al. [126] proved that *M. pulcherrima* effectively reduced disease symptoms caused by four strains of *P. expansum*. Alvarez et al. [127] also demonstrated the efficacy of *Candida sake* (Saito & Oda) van Uden & H.R. Buckley isolated from the Arctic environment against *P. expansum*. Sun et al. [121] showed that the cell wall of *Rhodosporidium paludigenum* (Fell & Tallman) Q.M. Wang, F.Y. Bai, M. Groenew. & Boekhout induced a strong immune response against *P. expansum* and Hershkovitz et al. [128] reported an enhanced immune response against *Penicillium digitatum* after treatment with a *Metschnikowia fructicola*-based preparation.

## 6. Yeast-Based Crop-Protection Products Available Worldwide

Several bioproducts based on yeast strains and one based on a substance derived from yeast cell walls are registered internationally at the moment (Table 3). Blossom Protect (fungicide and bactericide), Botector and BoniProtect (fungicides) contain germinated cells of *A. pullulans* (strains DSM 14940 and DSM 14941). Blossom Protect is intended for use against fire blight, bitter rot, grey mold, wet and brown rot and anthracnose in fruit storage and apple orchards. Botector prevents grey mold on grapevines, strawberries and other fruit. BoniProtect is used against fungal diseases caused by *Pezicula* sp., *Nectria* sp., *B. cinerea*, *Monilinia fructigena* Honey and *P. expansum* in orchards. Julietta is a fungicide containing the LAS02 strain of S. cerevisiae, designed to prevent grey mold on strawberries and tomatoes in greenhouses and under covers. Nexy contains the yeast *C. oleophila* and is used against grey and blue mold in apple and pear fruit storage. Noli, containing *Metschnikowia fructicola* strain NRRL Y-27328 KM1110 WDG, is used against postharvest decay in certain fruits and berries caused by *Botrytis* and *Monilinia* spp. Romeo, a product

containing cerewisan, and whose main ingredient is the cell walls of S. cerevisiae, is used to prevent powdery mildew and grey mold on crops such as grapevines, lettuce, tomato, strawberry and cucumber.

**Table 3.** Commercially available yeast-based bioproducts for the control of plant diseases.

| Yeast | Product Trade Name | Target Pathogens | Crops |
|---|---|---|---|
| *A. pullulans* | Blossom Protect | *E. amylovora*, *B. cinerea*, *Colletotrichum gloeosporioides* | Apples |
| *A. pullulans* | Botector | *B. cinerea* | Apples, pears, grapevines, strawberries and other fruit |
| *A. pullulans* | BoniProtect | *Pezicula* sp., *Nectria* sp., *B. cinerea*, *M. fructigena*, *P. expansum* | Apples, pears |
| *C. oleophila* | Nexy | *P. expansum*, *B. cinerea* | Apples, pears |
| *M. fructicola* | Noli | *B. cinerea*, *Monilinia* spp. | Soft fruit (including strawberry), stone fruit and table and wine grapes |
| *S. cerevisiae* (cell walls) | Romeo | *B. cinerea*, *Erysiphales* | Grapevines, lettuce, tomato, strawberry and cucumber |

More yeast-based products are also commercially available as plant development and growth enhancers, often combined with other microorganisms and plant extracts. Although they are available on the market on the basis of legal acts on fertilization, their agricultural suitability is not subject to such rigorous assessment, as is the case with plant protection products. It is also difficult to compile a list of such commercial products available in different countries.

### 7. Challenges and Possibilities for Yeast-Based Bioproducts

Incorporating microbiological agents in plant protection can help to minimize or exclude the use of agrichemicals and enhance plant quality. However, biological plant protection products need to meet strict criteria. They need to exhibit high pathogen inhibition ability. Development and implementation take many years for both in vitro and in planta studies, and are expensive. While their biomass production should be cost-effective, the process is usually complicated, and time and resources are intensive. A proper carrier (such as lignite dust) needs to be used to ensure the microorganisms' survival in the environment. Proper formulation is key for their usability and viability. The right carrier, which is effective, biodegradable and nonpolluting, can increase the biocontrol agent's efficiency and lifespan, including yeasts. Among the types of formulations for biopesticides, presently solid (peat, powder and granules) and liquid carriers are used. It is also crucial that the antagonistic properties proved in the laboratory are preserved irrespective of the production scale. Another issue is maintaining their antagonistic properties and ensuring their consistency of performance under different environmental conditions. It is also important to consider the microorganisms' compatibility with the plant. These challenges are some of the reasons why the number of yeast strains exhibiting antagonistic activity against plant pathogens in laboratory experiments is much higher than those implemented into practice. To overcome those issues, systemic biocontrol strategies need to be developed, that take into consideration beneficial microorganisms, crops, pests and agricultural practices alike [129]. Additionally, the current productive structure, including technical production systems, regulations and markets, needs to be adjusted to be suitable for biocontrol methods and strategies [130].

However, despite these challenges, there is a need for biological organic crop protectants and yield enhancers to be developed and put into practical use. In addition, yeasts can be used not only as biocontrol agents against plant pathogens, but as mentioned above, they are environmentally safe and can participate in the bioremediation strategy [131].

*Rhodotorula glutinis* and *Rhodotorula rubra* have been shown to degrade organophosphorus chlorpyrifos [132] and *Rhodotorula mucilaginosa* was used to eliminate neonicotinoid insecticides and thiacloprid [72].

Genetically modified yeast strains *M. pulcherrima*, *Cryptococcus tephrensis* and *A. pullulans* can also reduce pest populations [133,134] in combination with granulovirus, which increases the mortality of the larvae and guarantees better protection of the apple tree against apple fruit invasion by *Cydia pomonella* [135] by producing pheromone components or precursors. Modified *Yarrowia liplytica* yeast produces the sexual pheromone of *Helicoverpa armigera*, effectively eliminating this pest in field experiments on cotton, tomato and corn [136–138].

## 8. Conclusions

Biological control with microorganisms is effective, sustainable and environmentally friendly [139]. Applying it can successfully reduce the need for chemical fungicides, whose harmful influence on human health and the environment is substantial [140]. In an age of increasing demand for biological plant-protection agents, further research leading to achieving this goal is necessary, especially considering that these living microorganisms need adequate conditions to survive after application, and thus the strategy of treatments based on living yeasts or substances produced by them must be developed together with the technology of production for these biological products [141,142].

The possibilities of using yeasts are very promising, more so as they have been known for several years as organisms with great protective potential. Studies are still being carried out to understand the biochemical mechanism between the plant pathogen and the yeast cell, as a result of which the plant's defense system is activated. Plants are known to produce the hormone abscisic acid (ABA) in response to abiotic stresses. The ABA signaling pathway is very complex and relies on a large number of copies of genes encoding homologous signaling components. Abscisic acid (ABA) is the main phytohormone involved in many developmental processes and in increasing resistance to environmental stresses. Yeast is used as a reconstitution system to investigate the functionality of this complex and the highly multiplexed core signaling pathway. Further work will be needed to investigate which new models derived from the reconstruction of the ABA signal transduction pathway in yeast reflect the signaling mechanisms present in plants. Perhaps it will help to isolate and start working with molecules and to develop new protective innovative bioproducts, as was in the case with nanocompounds [143]. The positive effect of agriculture-friendly nanocompounds, sometimes combined with bioinoculants, that can be used as a good alternative to chemical fertilizers in sustainable agriculture was confirmed [144,145]. Now is a new challenge to develop nanotechnology with yeast molecules to be used for plant protection.

**Author Contributions:** Conceptualization, J.K. (Jolanta Kowalska) and J.K. (Joanna Krzymińskaand); investigation, J.K. (Jolanta Kowalska) and J.K. (Joanna Krzymińskaand); writing—original draft preparation and editing, J.K. (Joanna Krzymińskaand), J.K. (Jolanta Kowalska) and J.T.; supervision, J.K. (Jolanta Kowalska) and J.T. All authors have read and agreed to the published version of the manuscript.

**Funding:** This research received no external funding.

**Institutional Review Board Statement:** Not applicable.

**Informed Consent Statement:** Not applicable.

**Data Availability Statement:** Not applicable.

**Conflicts of Interest:** The authors declare no conflict of interest.

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
