# Peer review of "Yeasts as a Potential Biological Agent in Plant Disease Protection and Yield Improvement—A Short Review"

_agriculture, doi:10.3390/agriculture12091404_

Round 1

Reviewer 1 Report

The authors made a review about the role of yeasts as biological control agents against diseases. the review is interesting but the information given by the authors, is not new since many of the compounds are very known that have beneficial effects in several crops, such as auxins, ethylene...etc.

The section 2 is enough known to add in a review therefore I suggest to delete the section keeping some general characteristics  but in  introduction section not as section 2.

The authors must to put attention to italize all microorganisms names (line 81, 118, 120-122, 166...)

The authors have included authors names after genera and species to specify the strain but this practice is uncomfortable and makes reading very difficult such as lines 118-122, 134-137, 155-156, 190-193, 209, 239-245 and so on.

Author Response

Dear Reviewer  1

We thank the reviewer for any suggestions to improve the manuscript. All suggested changes important for the improvement and preparation of a more interesting and valuable manuscript have been included. The authors declare that the manuscript has been checked and corrected by a native speaker, Dr. Robert Lee from the University of Warmia and Mazury in Olsztyn, prior to its submission. The authors believe that the language is appropriate for scientific work. Detailed responses for Reviewer are in italic and are provided below.

  1. The authors made a review about the role of yeasts as biological control agents against diseases. the review is interesting but the information given by the authors, is not new since many of the compounds are very known that have beneficial effects in several crops, such as auxins, ethylene...etc. - indeed it is so, but in order to comprehensively present the topic, the text includes well-known scientific papers, but also new and latest ones.
  2.  The section 2 is enough known to add in a review therefore I suggest to delete the section keeping some general characteristics  but in  introduction section not as section 2. – it was corrected according to comment of Reviewer.
  3.  The authors must to put attention to italize all microorganisms names             (line 81, 118, 120-122, 166...) – it was checked and done.
  4. The authors have included authors names after genera and species to            specify the strain but this practice is uncomfortable and makes reading          very difficult such as lines 118-122, 134-137, 155-156, 190-193, 209, 239-       245 and so on. – yes, we are agree that it is difficult to reading but it is in        line with Taxonomy Browser. Authors propose to keep this manner of typing.

Reviewer 2 Report

The manuscript entitled “Yeasts as a Promising Biological Agent to Protect Plants and 2 Yield against Diseases – a Short Review” is interesting in field of agriculture as biocontrol agent. Some minor corrections are required.

1.     Explain one paragraph related to the other microbes which improved plant growth and use as biocontrol agent and how yeast is better than others. Authors can see these papers or other recent references. doi: 10.3389/fenvs.2021.769871, https://doi.org/10.1007/s40009-021-01047-w, https://doi.org/10.1007/s11104-022-05351-2.

2.     Is there signalling pathway between host plant and yeast to improve plant growth/ biocontrol agent. Explain in details.

Author Response

Dear Reviewer, 

We thank the reviewer for any suggestions to improve the manuscript. All suggested changes important for the improvement and preparation of a more interesting and valuable manuscript have been included. The authors declare that the manuscript has been checked and corrected by a native speaker, Dr. Robert Lee from the University of Warmia and Mazury in Olsztyn, prior to its submission. The authors believe that the language is appropriate for scientific work. Detailed responses for Reviewer are in italic and are provided below.

  1. Explain one paragraph related to the other microbes which improved plant growth and use as biocontrol agent and how yeast is better than others – additional general papers on biological agents (No. of ref. 129,130) and prospects for the development of the biological protection sector and stimulation of plants development by microbes have been added (ref. 137-145). The authors do not declare that yeast is better than other beneficial microorganisms, but only want to present a wide range of their possibilities in practice.
  2. Authors can see these papers or other recent references. – two from three proposed papers are added (ref. 144, 145)
  3. Is there signalling pathway between host plant and yeast to improve plant growth/ biocontrol agent. Explain in details – in 3. part of the manuscript is described this topic, so we do not think to develop even more it in details. Our manuscript is planned as short review.  The work is more focused on protection with the use of yeast than on explaining molecular, biochemical and genetic issues - these are scientific areas that the authors do not deal with. However, one more paper has been added, expanding a bit about signaling pathway between plants and yeasts (143).

Reviewer 3 Report

This is an interesting study upon “Yeasts as a Promising Biological Agent to Protect Plants and Yield against Diseases – a Short Review,” and the authors have collected a unique dataset using the cutting-edge methodology. The paper is generally well written and structured. However, in my opinion, the paper has some shortcomings in regards to some data analyses and text, and I feel this unique dataset has not been utilized to its full extent. Below I have provided numerous remarks on the text as it is often vague and long-winded. In several instances, I also suggested citing more relevant and recent literature. Furthermore, I have made line by line additional suggestions for more in-depth analyses of the data. Key critical points are:-

Title: Please change the title to “Yeasts as a potential biological agent in plant disease protection and yield improvement - a short review”

Abstract

Please follow the below-mentioned comments to improve the quality of the paper:-

Line No. 13-14: Rewrite the sentence to “Yeasts are microorganisms that live in a variety of environments and are antagonists to a wide range of plant pathogens.”

Line No. 17: Use the plural “characteristics” instead of “characteristic”

Line No. 20-24: Some of the topics described in detail are not mentioned in abstract

Introduction

Line No. 30: Add a hyphen in “microbial-based”

Line No. 59: Remove the comma after the word ecosystems

Line No. 61: Change the word “easy” to “easily”

Biological characteristics of yeasts

Line No. 65: Add a comma after “i.e.”

Line No. 73: Use the word “either” after “They are”

Line No. 75: Add a hyphen in a single cell

Line No. 76: Remove the word “on”

Bioactivity mechanisms of yeast

Line No. 99: Add a comma after agents

Line No. 106: Replace “enhances” with “enhance”

Line No. 125: Add a comma after 2016

Line No. 126: Replace “In accordance with” with “under” or “per”

Line No. 129: Use singular “yeast” in place of yeasts

Line No. 179-184: Effect of yeasts on plant immunity should be more detailed

Conclusion

Line No. 383: replace “demands” with “demand”

Line No. 381-387: No remedy for challenges faced in field application of yeast as a biocontrol agent is described

References

Try to use the latest references. The maximum references you coded in this article are old.

Similarly, I will suggest taking help from any English native speaker for the improvement of English write-up, grammar, punctuation, and sentence structure. And please provide a certificate from the agency or person for the same.

Keeping in view all the above, after thorough consideration, I will suggest minor revisions.

Author Response

Dear Reviewer, 

We thank the reviewer for any suggestions to improve the manuscript. All suggested changes important for the improvement and preparation of a more interesting and valuable manuscript have been included. The authors declare that the manuscript has been checked and corrected by a native speaker, Dr. Robert Lee from the University of Warmia and Mazury in Olsztyn, prior to its submission. The authors believe that the language is appropriate for scientific work. Additional general papers on biological agents and prospects for the development of the biological protection sector and stimulation of plants development by microbes have been added (129,130, 137-145). Detailed responses for Reviewer are in italic and are provided below.

  1. Title: Please change the title to “Yeasts as a potential biological agent in plant disease protection and yield improvement - a short review”- it was changed
  2. Abstract – all changes were made

Line No. 13-14: Rewrite the sentence to “Yeasts are microorganisms that live in a variety of environments and are antagonists to a wide range of plant pathogens.” – it has been rewritten

Line No. 17: Use the plural “characteristics” instead of “characteristic” – it has been corrected

Line No. 20-24: Some of the topics described in detail are not mentioned in abstract – The authors declare that the most important topics described in the manuscript are mentioned in Abstract, but we apologize in advance if any issues are omitted.

  1. Introduction - all changes were made

Line No. 30: Add a hyphen in “microbial-based” - it  has been corrected

Line No. 59: Remove the comma after the word ecosystems - it has been corrected

Line No. 61: Change the word “easy” to “easily” - it  has been corrected

  1. Biological characteristics of yeasts - changes were made

Line No. 65: Add a comma after “i.e.” - it  has been corrected

Line No. 73: Use the word “either” after “They are” - it  has been corrected

Line No. 75: Add a hyphen in a single cel - it  has been corrected

Line No. 76: Remove the word “on” - it  has been corrected

  1. Bioactivity mechanisms of yeast- changes were made

Line No. 99: Add a comma after agents – it has been corrected

Line No. 106: Replace “enhances” with “enhance” – it has been corrected

Line No. 125: Add a comma after 2016 – it has been corrected

Line No. 126: Replace “In accordance with” with “under” or “per” – it has been corrected

Line No. 129: Use singular “yeast” in place of yeasts – it has been corrected

       Line No. 179-184: Effect of yeasts on plant immunity should be more detailed – this topic is described in part no. 3 of the manuscript.   The work is more focused on protection with the use of yeast than on explaining molecular, biochemical issues – additionally these are scientific areas that the authors do not deal with. However, one more paper has been added, expanding a bit about signaling pathway between plants and yeasts (143).

  1. Conclusion - changes were made

Line No. 383: replace “demands” with “demand” – it has been corrected

Line No. 381-387: No remedy for challenges faced in field application of yeast as a biocontrol agent is described – more details and papers are added (141-145)

  1. References

Try to use the latest references. The maximum references you coded in this article are old.- The authors prepared this manuscript based on papers from a wide range of publication years, many of them are published 5 years ago, so it is difficult to say that these works are too old. Moreover, in order to comprehensively present the topic, the text includes well-known scientific works, however some a new papers are also added to revised manuscript  - the newest papers (ref. 129-130) and  ref. 137-145.

  Similarly, I will suggest taking help from any English native speaker for the improvement of English write-up, grammar, punctuation, and sentence structure. And please provide a certificate from the agency or person for the same - Authors declare that the manuscript before submission was checked and corrected by native speaker, Dr. Robert  Lee from the University of Warmia and Mazury in Olsztyn. Authors believe that the language is in appropriate manner for scientific papers. Mr. Dr. Robert Lee cooperates with us during the preparation of our scientific papers and they were always well prepared linguistically and stylistically.